# Membrane Progesterone Receptors (mPRs/PAQRs) Are Going beyond Its Initial Definitions

**DOI:** 10.3390/membranes13030260

**Published:** 2023-02-22

**Authors:** Justin Aickareth, Majd Hawwar, Nickolas Sanchez, Revathi Gnanasekaran, Jun Zhang

**Affiliations:** Department of Molecular and Translational Medicine (MTM), Texas Tech University Health Science Center, El Paso, TX 79905, USA

**Keywords:** progesterone (PRG), CmPn signaling network, CmP signaling network, CCM signaling complex (CSC), classic nuclear progesterone receptors (nPRs), non-classic membrane progesterone receptors (mPRs/PAQRs), genomic PRG actions, non-genomic PRG actions

## Abstract

Progesterone (PRG) is a key cyclical reproductive hormone that has a significant impact on female organs in vertebrates. It is mainly produced by the corpus luteum of the ovaries, but can also be generated from other sources such as the adrenal cortex, Leydig cells of the testes and neuronal and glial cells. PRG has wide-ranging physiological effects, including impacts on metabolic systems, central nervous systems and reproductive systems in both genders. It was first purified as an ovarian steroid with hormonal function for pregnancy, and is known to play a role in pro-gestational proliferation during pregnancy. The main function of PRG is exerted through its binding to progesterone receptors (nPRs, mPRs/PAQRs) to evoke cellular responses through genomic or non-genomic signaling cascades. Most of the existing research on PRG focuses on classic PRG-nPR-paired actions such as nuclear transcriptional factors, but new evidence suggests that PRG also exerts a wide range of PRG actions through non-classic membrane PRG receptors, which can be divided into two sub-classes: mPRs/PAQRs and PGRMCs. The review will concentrate on recently found non-classical membrane progesterone receptors (mainly mPRs/PAQRs) and speculate their connections, utilizing the present comprehension of progesterone receptors.

## 1. Introduction

As one of the most important sex steroids and key cyclical reproductive hormones with major impacts on the female organs of vertebrate species, progesterone (PRG) is mainly produced by the corpus luteum (CL) of the ovaries. Besides its reproductive significance, PRG can also be generated from the adrenal cortex, Leydig cells of the male testes, adipose cells and neuronal and glial cells [1,2,3,4,5,6], reaffirming its wide range of physiological effects, especially in the metabolic systems, central nervous systems and reproductive systems in both genders [7,8,9,10,11,12,13,14,15,16]. PRG was first purified almost a century ago as an ovarian steroid with hormonal function for pregnancy [8,17]. Since its identification, the main function of PRG was quickly realized as pro-gestational proliferation during pregnancy [7]. However, how PRG exerts its biological functions remained a mystery until the first set of PRG receptors, classic nuclear progesterone receptors (nPRs), were identified 40 years ago [18,19,20,21]. nPRs have two main alternatively spliced isoforms, nPR-A/1 and nPR-B/2, as ligand PRG-bound transcription factors with intricately reciprocal regulation. nPR-B acts as a ligand-bound transcription activator of PRG-responsive genes, while nPR-A can act as an nPR-B antagonist [22,23]. Interestingly, vascular expression of nPR-A isoform is greater in women than in men, suggesting the gender influence of PRG actions in vascular angiogenesis [24]. Tremendous efforts have been made for the physiological functions of the canonical genomic pathway after binding of PRG to nPRs in classic genomic actions [25]. 20 years after the discovery of nPRs, PRG was also found to evoke rapid membrane and cytoplasm changes, defined as rapid, non-classic or non-genomic actions [26,27,28], opening a new venue for our understanding of the physiological functions of PRG-mediated actions. Currently, most results are generated from classic PRG-nPRs signaling cascades in the reproductive system due to the nPRs’ much earlier discovery [7,8,9,11,12].

In sum, PRG elicits its cellular effects by means of balanced signaling pathways that involve classic, non-classic or blended responses. The PRG receptors are responsible for binding to the molecule. A recent study has demonstrated a blended response through nPRs and mPRs, where mPRα plays a role in the short-term inhibition of PRG on focal adhesion signaling in endothelial cells, while nPR has long-term inhibitory effects [29]. However, accumulated data have been skewed to well-documented classic PRG-nPR-paired actions as nuclear transcriptional factors, which can be auto-regulated through the opposite effects of nPR-alternative spliced isoforms (PR-A/B, 1/2) [25]. Finally, it needs to be emphasized that PRG has been defined to be able to evoke both genomic and non-genomic PRG actions, solely through classic nPRs [26,27]. Intriguingly, new evidence also indicates that PRG likely exerts a wide range of PRG actions through non-classic mPRs/PAQRs, suggesting that both classic nPRs and non-classic mPRs/PAQRs can independently exert both their genomic and non-genomic PRG actions in parallel, which will be the focus of this review.

## 2. Discovery of mPRs/PAQRs Enriched Our Knowledge of PRG Signaling

The discovery of PRG was followed shortly by the discovery of its ability to mediate non-classical or non-genomic actions at the cell membrane [30,31]. However, it took another 20 years for the discovery of a membrane-bound PRG receptor called mPR [32,33,34,35].

mPRs are structurally different from the classic PRG receptors [36], and can be divided into two sub-classes: Class II PRG receptors (mPRα, β, γ, φ, ε) and adipoQ receptors (PAQR5, 6, 7, 8, 9), and the second sub-class, membrane-associated PRG receptor (MAPR). Four MAPR members have been identified: PGRMC1/2, neudesin (NENF) and neuferricin (NEUFC) [35,36,37,38]. There is debate on whether PGRMC1 and S2R work together to perform the same function [39,40,41,42,43]. The classification and cellular mechanism of membrane PRG receptors are still not well defined; PGRMC1 only has been proven to have moderate binding affinity for PRG [44]. Initially, mPRs were found to be highly expressed in reproductive tissues and some cancers [45,46,47], and it was believed that they function by coupling to G-proteins for rapid non-genomic PRG actions [48,49]. The first attempt to create homozygous mPR knockout mutant fish resulted in no observable changes in phenotype, implying the presence of alternative mPRs that can compensate [50]. Nonetheless, later findings suggest that the elimination of distinct membrane PRG receptors can have varying impacts on the processes of oocyte maturation and ovulation [51,52].

## 3. mPRs Play an Important Role in PRG Biogenesis and PRG-Mediated Signaling

### 3.1. mPRs Are Key Factors in the Mammalian Female Reproductive System

Classic nPRs were originally identified in the female reproductive system [18,19,20,21]. Its impacts on the female reproductive system and related cancers have been the main focus of research since their discovery [10,11,12,53,54]. Similarly, “non-classic” mPRs were also initially identified in the ovaries of female vertebrates [32,33,55]. Therefore, initial works of mPRs also emphasized reproductive organs, tissues and related cancers [56,57,58,59]. Following our deeper understanding of mPRs, however, the current focus has extended to many other tissues and various other cancers [36,60,61], especially in the central nervous system (CNS) [62,63,64]. Although there are many similarities between nPRs and mPRs regarding their PRG signaling actions, many obvious differences between these two types of PRG receptors have also started to be defined, as described below.

### 3.2. Different PRG Binding Mechanism between mPRs and nPRs

Recently, the PRG-binding pocket of mPRα has been put through a combination of homology modeling [65], utilizing the X-ray crystallographic structure of a related class of PAQRs, the AdipoRs [66]. The crystal structures of two subtypes of the AdipoRs, ADIPOR1 and ADIPOR2, were analyzed comparatively, revealing that they possess a similar architecture comprised of seven transmembrane domains and a large internal cavity capable of accommodating a free fatty acid (FFA). Additionally, they contain a zinc-binding site near the intracellular surface, within the seven transmembrane domains [66]. The homology models of mPRα were built based on the AdipoRs structure (which shares 26% sequence identity with mPRα) [65]. These models were confirmed through mutational studies and binding experiments with PRGs. The results showed that the formation of strong hydrogen bonds between the glutamine residue at 206 (Q206) in the binding pocket and the PRG would be disrupted if the positively charged arginine (R) was replaced, highlighting the key role of Q206 in forming these bonds. The positive charge of amino acids like arginine and lysine (R, K) can draw polar steroids into the hydrophobic progesterone binding pocket of mPRα, hindering PRG binding. However, the model predicts that the presence of a Zinc binding domain in the binding pocket could be a potential solution by forming a salt with the carboxylate group of the polar steroids and preventing its binding to the pocket. This prediction was supported by experiments which demonstrated that the binding of H3-isotope-labeled PRG to the mPRα with the Q206 mutant in the PRG-binding pocket was restored in the presence of 100 μM zinc [65]. Additionally, D-e-MAPP, a ceramidase inhibitor, was found to effectively block PRG-mPRα signaling by binding to the PRG pocket, suggesting its potential as an mPR-specific antagonist [65]. Additional data showed that unlike nPRs, the 3-deoxysteroids had similar affinities to their corresponding 3-keto analogs, indicating that mPRs do not prefer to bind to the 3-keto group [65,67]; however, the 3-keto group is essential for PRG binding to the nPRs [68]. These results support the notion that the PRG binding mechanism between mPRs and nPRs is indeed different [65,67,68,69].

### 3.3. Different Oligomerizing Mechanism between nPRs and mPRs for Ligand-Binding

As ligand-dependent transcription factors, nPRs form dimers to bind to specific palindromic PRG response elements (PRE) in the DNA promoter region, in order to activate PRG-responsive gene transcription after binding to PRG [70,71]. Utilizing two nPR counteractive isoforms, nPR-A and nPR-B, PRG action can be diversified by the combined actions of nPR-A and nPR-B, which alternatively form homodimers upon ligand binding (PRG-bound nPR-A/nPR-A and nPR-B/nPR-B) or heterodimers (PRG-bound nPR-A/nPR-B) that have distinct transcriptional activities at specific sets of gene promoters [71,72]. Similarly, nPRs can also bind their antagonists (such as ZK98299/Onapristone) or PRG receptor modulators (PRMs) to either impair PRG-nPRs binding to DNA (type-I) or form a dimer to bind to specific PREs in the promoter region to inhibit specific sets of PRG-responsive gene transcription (type-II, such as RU486/Mifepristone) [73,74,75]. Furthermore, as an antagonist of nPRs, mifepristone (MIF/RU486) can bind to nPRs to form either heterodimers (nPR-PRG/nPR-MIF), which have no binding affinity to PRE; or homodimers (nPR-MIF/nPR-MIF), which have much higher affinity to PREs than their homodimer counterpart (nPR-PRG/nPR-PRG) for inhibition of classic PRG actions [76,77,78].

Interestingly, protein structural studies of mPRs suggest that hydrophobic heme–heme stacking between PGRMC1/2 proteins lead to the formation of either homodimers (PGRMC1) [79] or heterodimers (PGRMC1/2) [80]. This dimerization of PGRMC1/2 proteins could result in their recruitment of mPRα (PAQR7) into the complex for PRG actions [80,81]. Despite the presence of compelling evidence that the formation of a complex between PGRMC1 and mPRα is necessary for PRG binding and its associated functions (Aizen, 2018 #432), the specific sequence of events leading to the formation of the PGRMC1/2 and mPRα (PAQR7) complex for PRG binding and PRG actions remains to be fully understood (Thomas, 2014 #436; Ventura-Bixenshpaner, 2018 #118; Peluso, 2022 #476).

### 3.4. Differential Pharmacodynamics of mPRs to Ligands Shared with nPRs

As an antagonist of PRG in the PRG-nPR signaling cascade, MIF certainly earned its candidacy early on as an initial therapeutic agent for the treatment of reproductive (breast, prostate, ovarian, endometrial) cancers and endometriosis, and is still on many active clinical trials [82,83,84,85]. Increased dosages of MIF with PRG can inhibit the growth of nPR(+/−) breast cancer cells and ovarian carcinoma cells, and induce apoptosis of these cells [86,87,88]. Clinically relevant doses of MIF significantly improve the efficacy of chemotherapy regimens for human ovarian carcinoma cells [89]. It has been found that MIF works in conjunction with high doses of PRG to inhibit endometrial cancer cell growth, and has known anti-cancer properties [87,88]. However, there have been conflicting results regarding whether MIF has growth inhibition or stimulation properties for hormone-dependent breast cancer cells [90,91,92]. The activity of MIF is dependent on specific cell types [89], MIF concentration and ratios of nPR isoforms [93]. MIF and PRG can act as negative regulators of proliferation in breast cancer cells [94,95] and as negative regulators of apoptosis to protect neuronal cells [96,97,98]. Additionally, as type-II selective progesterone receptor modulators (SPRMs), MIF has been found to work with JDP2 to activate the NTD of nPR as a nPR agonist [99,100]. By screening various cancer cell lines with different genetic backgrounds regardless of tissue origin, hormone responsiveness or competitively binding to mPRα, PRG and MIF can act as negative regulators of proliferation for controlling breast cancer cells [94,95] or negative regulators of apoptosis to protect neuronal cells [96,97]. Follow-up studies show that paired PRG-MIF has “synergistic” but “reversed” actions through mPRs; in contrast with PRG-nPR action, where MIF is antagonistic to nPR-PRG actions. This activity of synergistically paired PRG-MIF further expands from an anti-tumor role to other activities of various cell types/tissues, and is found to be solely mPR-dependent and unrelated to nPRs [94,95,96,97,101,102,103,104,105].

## 4. mPR-Mediated Signaling Can Be Coupled with Other Steroid Signaling Pathways

### 4.1. mPRs and PGRMC Can Form Their Own Complexes

mPRs and PGRMC1 are two newly identified sub-classes of PRG receptors that are widely expressed in various cells and tissues [106] and have been suspected to form a PRG membrane receptor complex to exert their PRG actions [80]. One study showed that ectopically expressed PGRMC1 in an nPR(−) breast cancer cell line (MDA-MB-231) can induce upregulation of both PGRMC1 and mPRα proteins on the cell membrane while also increasing PRG levels bound to mPRα in cell membranes. This observation was confirmed in several nPR(−) breast cancer cell lines [107]. Similarly, loss of PGRMC1 leads to decreased protein expression levels of mPRα in PGRMC1-knockout zebrafish [108], further supporting the existence of the aforementioned PGRMC1/2-mPRα complex, as well as the complex’s influence on the levels of mPRα expression.

### 4.2. mPR-Mediated Signaling and *Ionotropic Neuronal Receptor* GABA_A_R Coupled with Their Common Ligands

It has been well defined that PRG and its metabolic derivatives (such as 3α, 5α-THPROG, allopregnanolone, pregnenolone, etc.) are neurosteroids that can influence the generation of action potentials through their interactions with neuronal membrane receptors [6,109,110,111,112]. One of the major targets of neurosteroids is the GABA-A receptor [6,112]. Neurosteroids can either positively or negatively regulate GABA-A receptor signaling, depending on the chemical properties of the specific neurosteroid [112,113,114,115]. Recent studies have supported the idea that neurosteroids act as agonists of mPR, and have a neuroprotective effect on neuronal cells [62,116,117]. This was confirmed by a recent study, which found that neurosteroids can only activate mPR-specific signaling in brain ECs that lack nPR and do not have GABA-A receptors [118]. These results indicate the important impacts of PRG-mPR actions on neuronal development, biogenesis and major functions [119,120].

### 4.3. Reciprocal Hormonal Regulation of mPRs and G-Protein Coupled Receptors

Previous reports suggest that mPRs are G-Protein Coupled Receptors (GPCRs), or are at least associated with G proteins to exert their cellular functions, such that mPRα, β and γ (PAQR7, 8, 5) are coupled with inhibitory G (Gi) proteins while both mPRφ and ε (PAQR6, 9) are coupled with stimulatory G (Gs) proteins [48,49,65,120,121,122,123]. However, later evidence suggests that mPRs belong to the adipoQ family (PAQRs) [48,124] instead of the GPCR class [37,125,126]. Interestingly, it was found that G protein-coupled estrogen receptor 1 (GPER) can coordinate with mPRs in a reciprocal fashion of hormonal regulation. In this hormonal feedback regulatory loop, PRG will increase expression of mPRs but decrease expression of GPER, in order to regulate the maturation of oocytes [127,128].

### 4.4. Crosstalk and Reciprocal Regulation between nPRs and mPRs

As mentioned previously, PRG is capable of exerting its cellular effects through either its classic, non-classic or combined responses through binding to either classic nPRs, non-classic mPRs or both simultaneously, warranting both pathways equally important status in PRG-mediated signaling. Both nPRs and mPRs can be coupled with other steroid signaling pathways [45,128,129,130,131,132,133,134], and evidence indicates the existence of the coupled mPR-nPR signaling cascade in nPR(+) cells [46,94,128,135,136]. Despite its significance, the relationship between classic and non-classic PRG receptors has been largely unexplored. It was reported that activation of mPRs can result in activation of nPRs, leading to a proposed model where steroid hormone-dependent mPRs contribute to later actions of nPR [137]. Recent evidence shows that CCM signaling complexes (CSC), consisting of CCM1, CCM2 and CCM3 proteins [138,139,140], can couple both nPR and mPR signaling cascades to bridge crosstalk among nPRs, mPRs and their shared ligands (such as PRG and MIF) to form the CSC-mPR-PRG-nPR (CmPn) signaling network in nPR(+) cells or the CSC-mPR-PRG (CmP) signaling network in nPR(−) cells [102,103,104,118,141]. This demonstrates that a common core mechanism exists, regardless of nPR(+/−) cell type. Chronic disruption of this intricate balance within the CmPn/CmP signaling networks, such as patients under HRT or females taking prolonged hormonal contraceptives, could result in perturbation of this signaling network with potential pathological consequences [118,141,142]. In human myometrial cells with high expression of nPR-B isoform, mPRα (PAQR7)-mediated signaling was found to be able to modulate nPR-B signaling to maintain the biogenesis of the myometrium [46,143]. This supporting evidence suggests that PRG signaling in nPR(−) cell lines might be mediated solely through mPRs (PAQRs) [94], a finding that was validated by recent data on the novel CSC-mPRs-PRG (CmP) signaling network in nPR(−) breast cancer cells [47]. Furthermore, with nPR(−) cells utilizing the CmP signaling network in most tissues and organs [118,141], the CSC is able to stabilize mPRs under mPR-specific PRG actions, indicating a more essential role of the CSC on the stability of mPRs (PAQRs) in nPR(−) cells. Our research revealed that the CSC can control the signaling pathways initiated by PRG receptors in breast cancer T47D cells [nPR(+)/mPR(+) and glucocorticoid receptor (−)]. The CSCs are able to link classic (nPR) and non-classic (mPR) signaling mechanisms, creating intercommunication between the two. Similarly, the actions of PRGs specific to mPRs have a reciprocal positive impact on protein expression in the CSC, which is comparable to the cellular compartments of other steroid receptors. Our findings indicate that under PRG stimulation, the stability of CSCs is regulated through two main signaling pathways: firstly, by the detrimental effects of PRG or its antagonist, mifepristone, which act through both types of PRG receptors; and secondly by the positive impact of nPR signaling. This highlights mPRs as a novel type of PRG receptor that functions similarly to traditional nPRs [105]. Our discovery highlights the significance of the balance between classic and non-classic PRG signaling in determining the function of the CSC. It also recognizes the CSC as a crucial mediator of cross-communication between nPRs and mPRs in cells that possess nPR. This observation is further strengthened by earlier findings that PRG can act on both nPRs and mPRs at the same time, and activating mPR signaling can enhance the hormone-activated nPR-2 isoform [46]. In summary, the intricate interplay between the CSC-mPRs-PRG-nPRs (CmPn) signaling network in T47D cells that have nPR can be understood as a universal mechanism present within the CmPn signaling network under steroidal influence, which have been validated in different nPR(−) breast cancer cells and nPR(−) ECs [47,102,103,105,118,142].

## 5. Both nPR and mPR Mediated Cellular Signaling Might Work in Parallel

### 5.1. mPR Subcellular Compartmentation

mPR expression patterns in many tissues and cell lines and the possible involvement of mPRs in various biological processes have been extensively studied, though contradicting data have been consistently presented for the subcellular compartmentation of mPRs [50,144]. So far, mPRs have been acknowledged as a novel category of PRG receptors, exhibiting only confirmed non-genomic PRG actions. The manner in which they are stimulated differs significantly based on the tissue and cell type, despite ongoing discussions about their subcellular compartmentation [144]. As previously mentioned, mPRs were first recognized as a new family of PRG receptors located on the cell membrane [32] and their signaling pathways and functions, triggered by PRG-mPRs, have been studied in various organisms and cell types [37,49,145,146,147]. Studies have revealed the cytoplasmic localization or compartmentation of mPRs in different cell types [121,148,149]. Initial pieces of evidence of cytoplasmic vesicles of mPRs [121,148] were speculated as the results of either technical issues or cytotoxic events for recombinant mPRs from transiently transfected constructs [148] or found in the membrane of the endoplasmic reticulum (ER) [121]. Contradictory evidence has surfaced regarding the localization of recombinant mPRα in CHO cells. Some studies have reported it in the ER, while others have found it in the nuclear membrane of small and large luteal cells [55]. Furthermore, recombinant mPRα has been observed in vesicles and the nuclear membrane in COS1 cells [150], and various forms of recombinant mPRs have been found in cytoplasmic vesicles and the ER in cells such as MB231 [121], HEK 293, LLC-PK and MDCK [151]. The debate continues on the cause of the cytoplasmic localization of mPRs. It is unclear whether this is due to transfection artifacts [148], clathrin-mediated endocytosis to cytoplasmic endosomes [149], the absence or insufficiency of PGRMC1 [107] or if mPRs were mistakenly classified as membrane proteins [144,150]. The latest subcellular compartmentation and cellular fractionation data indicate that mPRβ can be colocalized in the nucleus and the cytoplasm [103], which matches to previous data from subcellular localization of recombinant mPRs in transient transfection experiments among various cells. The nuclear compartmentation with strongly condensed localization within the nucleolus of recombinant NFlag-mPRα protein in HEK293 cells [151] and in MDCK cells [144] match recent findings of nuclear trafficking endogenous mPRβ in MB231 cells [103]. Furthermore, the nuclear localization of endogenous mPRα in M11 cells [149] and COS-1 cells [121] provide more supportive evidence. In summary, new evidence provides new insights on subcellular compartmentation of mPRs from its originally defined cytoplasmic membrane into cytosols, perinuclear region and intra-nucleus, indicating much wider cellular functions of mPRs than previously proposed.

### 5.2. Cytoplasmic-Nuclear Trafficking Is the Common Theme for Steroid Hormone Receptors

The most common and significant characteristic shared by all steroid hormone receptors is the ability for cytoplasmic-nuclear trafficking [152]. The molecular mechanisms of nucleocytoplasmic shuttling have been well defined for AR [153,154], GR [155], ERs [156] and nPRs [157]. A recent finding that nucleocytoplasmic shuttling of mPRβ (PAQR8), along with its newly defined nuclear exit signal (NES) and nuclear localization signal (NLS) within the amino acid sequence, is a dynamic event under mPR-specific PRG actions in nPR(−) triple negative breast cancer (TNBC) cells. This nucleocytoplasmic shuttling of mPRβ is almost identical to the behavior of other steroid hormone receptors [103]. These findings indicate that the currently classified non-classic mPRs are actually a novel group of PRG receptors with similar functions as classic nPRs that can potentially perform both genomic and non-genomic PRG-actions, with possibly more tissue- and cell-specific intracellular performance. Lastly, additional compelling proof can be found in the localization of PGRMC1, a sub-class of membrane progesterone receptors, in the nucleus. This occurs after treatment with human chorionic gonadotropin (hCG) in rat granulosa cells [158]. PGRMC1 displays a similar hormonal response to mPRs, exhibiting nucleocytoplasmic shuttling. Since then, PGRMC1 has been widely accepted as localizing to both the nucleus and the cytosol. So far, PGRMC1 cytoplasmic-nuclear trafficking has been observed in spontaneously immortalized granulosa cells (SIGCs) [10,159,160,161], human granulosa cells [160] and human ovarian cancer cells [10,80,158,160]. The evidence shows that mPRs can trigger fast, non-genomic actions on the cell surface [162]. However, growing data suggest that there may be multiple PRG-mPR signaling pathways leading to PRG actions. While there are no reports of DNA binding from the PRG-mPR complex yet, it is still unclear whether mPR can exert its genomic actions by directly binding to the PRE (similar to nPR’s role as a transcription factor) or indirectly through interactions with other transcription factors or GPCRs. Further investigation is required to clarify these possibilities.

## 6. Key Roles of mPRs within the CmPn/CmP Signaling Network

Although the CSC was initially defined during CCM-related hemorrhagic stroke studies [138,139,140,163], its relationship with mPRs was just recently investigated [102,103,118]. The exact intracellular physiological mechanisms by which these hormones function remains largely unstudied. Recent evidence suggests that the CSC functions as a bridge for crosstalk among nPRs, mPRs and their ligands, to form what we proposed as the CSC-mPR-PRG-nPR (CmPn)/CSC-mPR-PRG (CmP) signaling networks, in nPR(+/−) cells, respectively [102,103,104,139,142,163,164]. Studies have shown variable expression patterns of mediators of this pathway across various breast cancers, suggesting this pathway is largely involved in breast tumorigenesis [102,103,163]. Some breast cancer subtypes expressing mPRs have partially contributed to the discovery of mPR-specific PRG actions of the CmP network. Furthermore, the findings of other cellular factors modulating the effects of PRG, mPRs, nPRs and the nucleocytoplasmic shuttling and localization properties of mPRs all call for further investigation into the CmPn/CmP signaling networks [141].

## 7. Conclusions

The cellular function of mPRs is a rather intricate topic related to specific forms of mPR, ratios of different subtypes, cellular compartmentations, nucleocytoplasmic shuttling dynamics and their relationship with nPRs—this may be the reason why so many contradicting results have been reported. In this review, we attempted to summarize newly defined cellular biogenesis, basic structural dynamics and its associated cellular functions of mPRs/PAQRs in an attempt to elaborate on the relationship between the classic nPRs and non-classic mPRs/PAQRs, as well as their response to shared ligands (PRG, MIF) (Figure 1). We provided evidences for newly defined cellular roles of mPR-mediated PRG actions in various cells, tissues and organs and the relationship among nPR(+/−) cell types, which might be responsible for the physiologic dynamic between mPRs/PAQRs and the CmPn/CmP signaling networks.

## Figures and Tables

**Figure 1 membranes-13-00260-f001:**
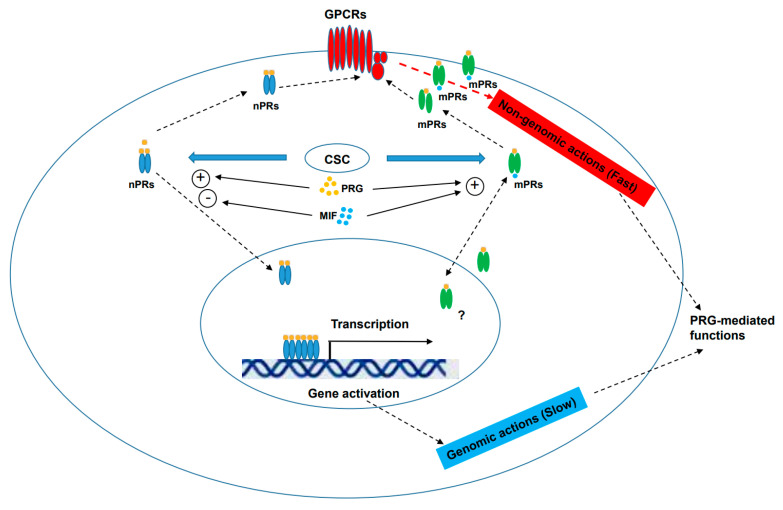
Signaling network bridge “crosstalk” among key players within CmPn signaling network. The diagram illustrates the relationship among PRG, nPR, mPR, CSC and the multiple ways PRG acts on classical nPRs and non-classical mPRs/PAQRs to influence cellular processes. We present that mPRs/PAQRs influence non-genomic actions, as well as possibly genomic actions, either through nPRs or independent of nPRs. Plus symbol indicates positive effects, while minus symbol imposes negative effects. In the figure, PRG, progesterone (gold-colored dot); MIF, mifepristone (light-blue-colored dots); nPRs (blue-colored), classic nuclear PRG receptors (blue-colored); mPRs, classic membrane PRG receptors (PAQRs, green-colored); CSC, CCM signaling complex; GPCRs: G-Protein Coupled Receptors (red-colored). Plus sign means positive effect, minus sign means negative effect and question mark indicates that the underlying mechanism is still unknown.

## Data Availability

Not applicable.

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
