# Peer review of "Membrane Progesterone Receptors (mPRs/PAQRs) Are Going beyond Its Initial Definitions"

_membranes, 2023, doi:10.3390/membranes13030260_

Round 1
Reviewer 1 Report
This manuscript is a short review on membrane progesterone receptors (mPRs) which have been recently shown to exert numerous new functions and therefore a topic worthy of a new review. The comparisons between nPRs and mPRs in the review are interesting. The discussion of the recent studies by the authors on CCM signaling complexes coupling to both nPR and mPRs signaling cascades is particularly interesting and the section on this topic should be expanded to discuss their findings in more detail. However, overall, the scholarship is inadequate in other sections of the review. The authors have an incomplete understanding of mPRs and the text has numerous errors and misconceptions which need to be corrected. In particular, there is an over reliance on citing broad reviews on the topic written by persons not directly involved with research in mPRs. Researching the primary literature on mPRs should address many of the misconceptions and mistakes in this manuscript. The lack of web addresses (DOI) for the cited papers greatly complicates the refereeing (157 references) so detailed comments on the appropriateness of the citations are limited to those in the first 100 lines of this review. The authors should check all the citations in the rest of the review. Throughout the review authors do not always distinguish between the mPR subtypes and describe which one is being referred to. This is very important for interpreting the findings. Also, the author’s results reviewed in this manuscript indicating progesterone actions mediated solely through mPRs did not describe the use of a specific mPR agonist (Org OD 02-0) or a nPR specific one such as R5020 which would have supported their conclusions. Throughout the manuscript the authors suggest MIF and PRG act together on mPRs, although no evidence is provided to support this assertion. See comments for figure 1 for a possible alternative interpretation. Finally, the authors make bold claims in this review of the functions on mPRs intracellularly in the absence of definitive data supporting their claims. It is strongly suggested that the authors tone down these claims and instead suggest them as hypotheses that require further examination and indicate what experiments could be conducted to test them
Review of first 100 lines
Title: The title is confusing. in some publications PGRMCs have been called mPRs e.g. Lin et al., 2015 Toxicology and Pharmacology 288 (3) 359-373. To avoid confusion, it is suggested that “(mPRs)” be replaced with “(mPRs, PAQRs)” Similarly, see the confusion in lines 68-73.
Abstract: the abstract is not very informative. The first part is repetitive and the abstract only briefly mentions the topic in the title “ are going beyond its initial definitions” It is recommended that more details on mPRs activating nPRs.
L49: The statement is incorrect and needs to be revised. It has been known since the late 1970s-1980s that progesterone acts at the cell membrane to mediate non-classical or non-genomic actions. e.g. Masui and Markert, 1971 J. Exp. Zool. 177) 129–145; Sadler and Maller 1982 J. Biol. Chem. 257 (1982) 355–361.
L52-53: This statement is unclear, what is being referred to with much earlier discovery? At this point the identities of membrane receptors have not been described, reference 29 appears to be inappropriate here, does not directly refer to this statement.
L56: here with reference 30 and elsewhere in this review the authors rely on reviews rather than the primary literature. This reference is especially inappropriate since it refers mainly to PGRMC1. The combined responses should be briefly listed here. For example, a recent example of a combined response through nPRs and mPRs : Pang and Thomas, 2022, J. Mol. Endocrinol. 70:1 e220073.
L 65: Title is incorrect see comment about line 49 above . It was known from early times that mPRs act independently to nPRs.
L68-73 and L81-82.:The term mPRs is confusing as discussed earlier for the title. Change mPRs to membrane progesterone receptors, the term mPRs should be reserved for PAQRs in this review.
L80: it should be clarified that PGRMC1 only has moderate binding affinity for PRG.
L84 Reference 44 there are better references than this such as 125,126
45. should include original reference for activating G proteins (reference 112)
L87-88: the statement is incorrect because the reference referred to here is a short meeting abstract with preliminary results which were inconclusive. A more complete study by Wu, Liu, Chen, Hong and Zhu 2020 Mol Cell Endocrinol. 511: 110856 shows that there are differential effects of knockout of the different mPRs on oocyte maturation and ovulation. The next needs to be revised accordingly.
L94: reference 34 is not appropriate here. These authors primarily discuss extranuclear signaling through the nPR. The paper by Ashley et al.,2006 Endocrinology 147 (9) 4145-4159 describes a sheep mPR and its localization in reproductive tissues including ovary.
L98: references 56,57 do not refer to mPRs in the CNS. More appropriate references are Meffre et al. 2013 Neuroscience 231: 111-124, Zuloaga et al 2012 Endocrinology 153:4432-443.
Abbreviated Review of rest of manuscript.
L104-109: The statement is unclear and needs to be revised. It is not clear that arginine is not in the binding pocket of wild type mPRα, only when its substituted at a key residue; this destroys binding which is only restored in the presence of zinc
L130-133: Again, the original reports on this are not included (reference 74?). Reference 98 should be quoted here.
L134-136: there is disagreement concerning this statement; there is clear evidence that formation of a PGRMC1 and mPRα complex is needed for PRG binding and PRG functions. Aizen et al. 2018. Gen .Comp. Endocrinol. 263, 51-61. Reference 99. This statement also conflicts with a later section of this review “mPRs and PGRMC1 can form complexes”
Section on differential dynamics… : The discussion on MIF with regard to mPRs is irrelevant , MIF does not active mPRs. This entire section discusses mainly nPRs and does not add any useful information on mPRs and the entire paragraph should be deleted.
L152-159: The text here is misleading “PRG and MIF can act as negative regulators of proliferation via mPRα” MIF does not bind or activate mPRα (reference 112). Reference 88 shows MIF does not act through mPRα.Reference 89 mainly refers to nPRs. Refer 90 is a general review and refence 91 refers to PGRMC1. Reference 92 is interesting with results in nPR-ve cell lines, but does not indicate it acts through mPRα. PGRMC1 and the GR are also potential targets. This needs to be clarified.
L157-160: there is no direct evidence to support the synergistic MIF-PRG action is “solely mPR-dependent”
L161-172: This section is incomplete. Reference 73 on granulosa cells supports this association. Also, another protein. Another mPR, beta has also been shown to associate with APPL and VDL ,Nader et la., Plos Biol 2021,19,e30001117.
L184: the results in reference 108 require clarification to confirm the statement that GABA A receptors are not present in the brain ECs.
L187-198: This section suffers from not referencing the primary literature that established early on that mPRs are not GPCRS and are PAQRs. Tang et al. 2005. J.Mol.Evol 61,372-380 showed 2 years after the discovery of mPRs that they are PAQRS and reference 112 showed 2 years later that GPCRs and mPRs have a different phylogenetic origin. The authors should replace recent references 958,109) with these and revise the text. There are no reports in the primary literature that mPRs are GPCRs. EST is not defined.
Crosstalk and reciprocal regulation ……
This section reports a novel mechanism which requires substantial elaboration on the role of CCM signaling complexes in mPR functions in order to convince the reader of their importance.
L203-204: add reference 118 (118-125
L204-205: the is no evidence of coupled mPR-PR in references 118, 128 (existence of nPR in T lymphocytes disputed)
L211: The statement “nPRs, mPRs, and their shared ligands (PRG, MIF)” is incorrect. There are no data showing that MF is a ligand for mPRs. It is frequently used as a negative control, to exclude actions through the nPR. This is a major problem that requires extensive explanation of the evidence supporting this claim.
L220: reference 135 refers to sperm, not the myometrium
mPR subcellular compartmentation
This section has numerous misleading statements and requires substantial modification. The discussion about earlier controversies is not relevant and dated. There is clear evidence that mPRs are localized in intracellular regions in addition to on the cell membrane and the authors describe the evidence in lines 256-267 supporting these other localizations which is not surprising, all membrane receptors also are present intracellularly.
L236-240. The beginning of this section is redundant since it well known that all membrane receptors are also present intracellularly where they a synthesized and internalized after activation. mPRα is rapidly internalized after ligand binding : Foster et al. 2010 Mol. Med. Rep. 3,27-35. Reference 46 is a very brief review and reference 136 is a commentary over a decade ago which is no longer relevant, especially the statement L239-240 that ”debates on their subcellular compartmentation continue”. This opinion was stated 13 years ago!.
L243: the statement : “recent evidence of cytoplasmic localization ….of mPRs…have emerged” is inaccurate because reference 110 refers to a study published in 2006, and reference 140 to a review published in 2008; not recent at all.
L250: Similarly, references 141 and 142 are not “recently contradictory evidence “
L255-256 the statement “or misclassification of mPRs as membrane proteins {136,142] are still under debate” is incorrect. These papers/ commentaries were published in 2010 and 2008 and have not been seriously considered recently because of the abundant evidence the mPRs are on the cell surface and are activated by PRG -BSA ligands that cannot enter into the cell.
L270- sentence incomplete “cytoplasmic-“
272-279 there is no reference to the study discussed here.
L281-286: PGRMC1 is not a close sibling of mPRs-completely different. Why is this evidence convincing that mPRs ac the same way? This text needs to be revised.
L286-290: this section is highly speculative and not supported by any data it is suggested that it be revised to reflect this
Figure 1:
1. What is GPCRs? mPRs are not GPCRs
2. mPR are definitely on the cell surface as well as intracellularly. mPR should be shown on the cell surface.
3. Here and elsewhere in the review MIF is shown to act on mPRs. If the authors suggest MIF does not have an inhibitory effect on mPRs like it does on nPRs then the results should be discussed as such and not that MIF and PRG act together on mPRs.
04-109: The statement is unclear and needs to be revised. It is not clear that arginine is not in the binding pocket of wild type mPRα, only when its substituted at a key residue; this destroys binding which is only restored in the presence of zinc
L130-133: Again, the original reports on this are not included (reference 74?). Reference 98 should be quoted here.
L134-136: there is disagreement concerning this statement; there is clear evidence that formation of a PGRMC1 and mPRα complex is needed for PRG binding and PRG functions. Aizen et al. 2018. Gen .Comp. Endocrinol. 263, 51-61. Reference 99. This statement also conflicts with a later section of this review “mPRs and PGRMC1 can form complexes”
Section on differential dynamics… : The discussion on MIF with regard to mPRs is irrelevant , MIF does not active mPRs. This entire section discusses mainly nPRs and does not add any useful information on mPRs and the entire paragraph should be deleted.
L152-159: The text here is misleading “PRG and MIF can act as negative regulators of proliferation via mPRα” MIF does not bind or activate mPRα (reference 112). Reference 88 shows MIF does not act through mPRα.Reference 89 mainly refers to nPRs. Refer 90 is a general review and refence 91 refers to PGRMC1. Reference 92 is interesting with results in nPR-ve cell lines, but does not indicate it acts through mPRα. PGRMC1 and the GR are also potential targets. This needs to be clarified.
L157-160: there is no direct evidence to support the synergistic MIF-PRG action is “solely mPR-dependent”
L161-172: This section is incomplete. Reference 73 on granulosa cells supports this association. Also, another protein. Another mPR, beta has also been shown to associate with APPL and VDL ,Nader et la., Plos Biol 2021,19,e30001117.
L184: the results in reference 108 require clarification to confirm the statement that GABA A receptors are not present in the brain ECs.
L187-198: This section suffers from not referencing the primary literature that established early on that mPRs are not GPCRS and are PAQRs. Tang et al. 2005. J.Mol.Evol 61,372-380 showed 2 years after the discovery of mPRs that they are PAQRS and reference 112 showed 2 years later that GPCRs and mPRs have a different phylogenetic origin. The authors should replace recent references 958,109) with these and revise the text. There are no reports in the primary literature that mPRs are GPCRs. EST is not defined.
Crosstalk and reciprocal regulation ……
This section reports a novel mechanism which requires substantial elaboration on the role of CCM signaling complexes in mPR functions in order to convince the reader of their importance.
L203-204: add reference 118 (118-125
L204-205: the is no evidence of coupled mPR-PR in references 118, 128 (existence of nPR in T lymphocytes disputed)
L211: The statement “nPRs, mPRs, and their shared ligands (PRG, MIF)” is incorrect. There are no data showing that MF is a ligand for mPRs. It is frequently used as a negative control, to exclude actions through the nPR. This is a major problem that requires extensive explanation of the evidence supporting this claim.
L220: reference 135 refers to sperm, not the myometrium
mPR subcellular compartmentation
This section has numerous misleading statements and requires substantial modification. The discussion about earlier controversies is not relevant and dated. There is clear evidence that mPRs are localized in intracellular regions in addition to on the cell membrane and the authors describe the evidence in lines 256-267 supporting these other localizations which is not surprising, all membrane receptors also are present intracellularly.
L236-240. The beginning of this section is redundant since it well known that all membrane receptors are also present intracellularly where they a synthesized and internalized after activation. mPRα is rapidly internalized after ligand binding : Foster et al. 2010 Mol. Med. Rep. 3,27-35. Reference 46 is a very brief review and reference 136 is a commentary over a decade ago which is no longer relevant, especially the statement L239-240 that ”debates on their subcellular compartmentation continue”. This opinion was stated 13 years ago!.
L243: the statement : “recent evidence of cytoplasmic localization ….of mPRs…have emerged” is inaccurate because reference 110 refers to a study published in 2006, and reference 140 to a review published in 2008; not recent at all.
L250: Similarly, references 141 and 142 are not “recently contradictory evidence “
L255-256 the statement “or misclassification of mPRs as membrane proteins {136,142] are still under debate” is incorrect. These papers/ commentaries were published in 2010 and 2008 and have not been seriously considered recently because of the abundant evidence the mPRs are on the cell surface and are activated by PRG -BSA ligands that cannot enter into the cell.
L270- sentence incomplete “cytoplasmic-“
272-279 there is no reference to the study discussed here.
L281-286: PGRMC1 is not a close sibling of mPRs-completely different. Why is this evidence convincing that mPRs ac the same way? This text needs to be revised.
L286-290: this section is highly speculative and not supported by any data it is suggested that it be revised to reflect this
Figure 1:
1. What is GPCRs? mPRs are not GPCRs
2. mPR are definitely on the cell surface as well as intracellularly. mPR should be shown on the cell surface.
3. Here and elsewhere in the review MIF is shown to act on mPRs. If the authors suggest MIF does not have an inhibitory effect on mPRs like it does on nPRs then the results should be discussed as such and not that MIF and PRG act together on mPRs.
This manuscript is a short review on membrane progesterone receptors (mPRs) which have been recently shown to exert numerous new functions and therefore a topic worthy of a new review. The comparisons between nPRs and mPRs in the review are interesting. The discussion of the recent studies by the authors on CCM signaling complexes coupling to both nPR and mPRs signaling cascades is particularly interesting and the section on this topic should be expanded to discuss their findings in more detail. However, overall, the scholarship is inadequate in other sections of the review. The authors have an incomplete understanding of mPRs and the text has numerous errors and misconceptions which need to be corrected. In particular, there is an over reliance on citing broad reviews on the topic written by persons not directly involved with research in mPRs. Researching the primary literature on mPRs should address many of the misconceptions and mistakes in this manuscript. The lack of web addresses (DOI) for the cited papers greatly complicates the refereeing (157 references) so detailed comments on the appropriateness of the citations are limited to those in the first 100 lines of this review. The authors should check all the citations in the rest of the review. Throughout the review authors do not always distinguish between the mPR subtypes and describe which one is being referred to. This is very important for interpreting the findings. Also, the author’s results reviewed in this manuscript indicating progesterone actions mediated solely through mPRs did not describe the use of a specific mPR agonist (Org OD 02-0) or a nPR specific one such as R5020 which would have supported their conclusions. Throughout the manuscript the authors suggest MIF and PRG act together on mPRs, although no evidence is provided to support this assertion. See comments for figure 1 for a possible alternative interpretation.. Finally, the authors make bold claims in this review of the functions on mPRs intracellularly in the absence of definitive data supporting their claims. It is strongly suggested that the authors tone down these claims and instead suggest them as hypotheses that require further examination and indicate what experiments could be conducted to test them
Review of first 100 lines
Title: The title is confusing. in some publications PGRMCs have been called mPRs e.g. Lin et al., 2015 Toxicology and Pharmacology 288 (3) 359-373. To avoid confusion, it is suggested that “(mPRs)” be replaced with “(mPRs, PAQRs)” Similarly, see the confusion in lines 68-73.
Abstract: the abstract is not very informative. The first part is repetitive and the abstract only briefly mentions the topic in the title “ are going beyond its initial definitions” It is recommended that more details on mPRs activating nPRs.
L49: The statement is incorrect and needs to be revised. It has been known since the late 1970s-1980s that progesterone acts at the cell membrane to mediate non-classical or non-genomic actions. e.g. Masui and Markert, 1971 J. Exp. Zool. 177) 129–145; Sadler and Maller 1982 J. Biol. Chem. 257 (1982) 355–361.
L52-53: This statement is unclear, what is being referred to with much earlier discovery? At this point the identities of membrane receptors have not been described, reference 29 appears to be inappropriate here, does not directly refer to this statement.
L56: here with reference 30 and elsewhere in this review the authors rely on reviews rather than the primary literature. This reference is especially inappropriate since it refers mainly to PGRMC1. The combined responses should be briefly listed here. For example, a recent example of a combined response through nPRs and mPRs : Pang and Thomas, 2022, J. Mol. Endocrinol. 70:1 e220073.
L 65: Title is incorrect see comment about line 49 above . It was known from early times that mPRs act independently to nPRs.
L68-73 and L81-82.:The term mPRs is confusing as discussed earlier for the title. Change mPRs to membrane progesterone receptors, the term mPRs should be reserved for PAQRs in this review.
L80: it should be clarified that PGRMC1 only has moderate binding affinity for PRG.
L84 Reference 44 there are better references than this such as 125,126
45. should include original reference for activating G proteins (reference 112)
L87-88: the statement is incorrect because the reference referred to here is a short meeting abstract with preliminary results which were inconclusive. A more complete study by Wu, Liu, Chen, Hong and Zhu 2020 Mol Cell Endocrinol. 511: 110856 shows that there are differential effects of knockout of the different mPRs on oocyte maturation and ovulation. The next needs to be revised accordingly.
L94: reference 34 is not appropriate here. These authors primarily discuss extranuclear signaling through the nPR. The paper by Ashley et al.,2006 Endocrinology 147 (9) 4145-4159 describes a sheep mPR and its localization in reproductive tissues including ovary.
L98: references 56,57 do not refer to mPRs in the CNS. More appropriate references are Meffre et al. 2013 Neuroscience 231: 111-124, Zuloaga et al 2012 Endocrinology 153:4432-443.
Abbreviated Review of rest of manuscript.
L104-109: The statement is unclear and needs to be revised. It is not clear that arginine is not in the binding pocket of wild type mPRα, only when its substituted at a key residue; this destroys binding which is only restored in the presence of zinc
L130-133: Again, the original reports on this are not included (reference 74?). Reference 98 should be quoted here.
L134-136: there is disagreement concerning this statement; there is clear evidence that formation of a PGRMC1 and mPRα complex is needed for PRG binding and PRG functions. Aizen et al. 2018. Gen .Comp. Endocrinol. 263, 51-61. Reference 99. This statement also conflicts with a later section of this review “mPRs and PGRMC1 can form complexes”
Section on differential dynamics… : The discussion on MIF with regard to mPRs is irrelevant , MIF does not active mPRs. This entire section discusses mainly nPRs and does not add any useful information on mPRs and the entire paragraph should be deleted.
L152-159: The text here is misleading “PRG and MIF can act as negative regulators of proliferation via mPRα” MIF does not bind or activate mPRα (reference 112). Reference 88 shows MIF does not act through mPRα.Reference 89 mainly refers to nPRs. Refer 90 is a general review and refence 91 refers to PGRMC1. Reference 92 is interesting with results in nPR-ve cell lines, but does not indicate it acts through mPRα. PGRMC1 and the GR are also potential targets. This needs to be clarified.
L157-160: there is no direct evidence to support the synergistic MIF-PRG action is “solely mPR-dependent”
L161-172: This section is incomplete. Reference 73 on granulosa cells supports this association. Also, another protein. Another mPR, beta has also been shown to associate with APPL and VDL ,Nader et la., Plos Biol 2021,19,e30001117.
L184: the results in reference 108 require clarification to confirm the statement that GABA A receptors are not present in the brain ECs.
L187-198: This section suffers from not referencing the primary literature that established early on that mPRs are not GPCRS and are PAQRs. Tang et al. 2005. J.Mol.Evol 61,372-380 showed 2 years after the discovery of mPRs that they are PAQRS and reference 112 showed 2 years later that GPCRs and mPRs have a different phylogenetic origin. The authors should replace recent references 958,109) with these and revise the text. There are no reports in the primary literature that mPRs are GPCRs. EST is not defined.
Crosstalk and reciprocal regulation ……
This section reports a novel mechanism which requires substantial elaboration on the role of CCM signaling complexes in mPR functions in order to convince the reader of their importance.
L203-204: add reference 118 (118-125
L204-205: the is no evidence of coupled mPR-PR in references 118, 128 (existence of nPR in T lymphocytes disputed)
L211: The statement “nPRs, mPRs, and their shared ligands (PRG, MIF)” is incorrect. There are no data showing that MF is a ligand for mPRs. It is frequently used as a negative control, to exclude actions through the nPR. This is a major problem that requires extensive explanation of the evidence supporting this claim.
L220: reference 135 refers to sperm, not the myometrium
mPR subcellular compartmentation
This section has numerous misleading statements and requires substantial modification. The discussion about earlier controversies is not relevant and dated. There is clear evidence that mPRs are localized in intracellular regions in addition to on the cell membrane and the authors describe the evidence in lines 256-267 supporting these other localizations which is not surprising, all membrane receptors also are present intracellularly.
L236-240. The beginning of this section is redundant since it well known that all membrane receptors are also present intracellularly where they a synthesized and internalized after activation. mPRα is rapidly internalized after ligand binding : Foster et al. 2010 Mol. Med. Rep. 3,27-35. Reference 46 is a very brief review and reference 136 is a commentary over a decade ago which is no longer relevant, especially the statement L239-240 that ”debates on their subcellular compartmentation continue”. This opinion was stated 13 years ago!.
L243: the statement : “recent evidence of cytoplasmic localization ….of mPRs…have emerged” is inaccurate because reference 110 refers to a study published in 2006, and reference 140 to a review published in 2008; not recent at all.
L250: Similarly, references 141 and 142 are not “recently contradictory evidence “
L255-256 the statement “or misclassification of mPRs as membrane proteins {136,142] are still under debate” is incorrect. These papers/ commentaries were published in 2010 and 2008 and have not been seriously considered recently because of the abundant evidence the mPRs are on the cell surface and are activated by PRG -BSA ligands that cannot enter into the cell.
L270- sentence incomplete “cytoplasmic-“
272-279 there is no reference to the study discussed here.
L281-286: PGRMC1 is not a close sibling of mPRs-completely different. Why is this evidence convincing that mPRs ac the same way? This text needs to be revised.
L286-290: this section is highly speculative and not supported by any data it is suggested that it be revised to reflect this
Figure 1:
1. What is GPCRs? mPRs are not GPCRs
2. mPR are definitely on the cell surface as well as intracellularly. mPR should be shown on the cell surface.
3. Here and elsewhere in the review MIF is shown to act on mPRs. If the authors suggest MIF does not have an inhibitory effect on mPRs like it does on nPRs then the results should be discussed as such and not that MIF and PRG act together on mPRs.
es are Meffre et al. 2013 Neuroscience 231: 111-124, Zuloaga et al 2012 Endocrinology 153:4432-443.
Abbreviated Review of rest of manuscript.
L104-109: The statement is unclear and needs to be revised. It is not clear that arginine is not in the binding pocket of wild type mPRα, only when its substituted at a key residue; this destroys binding which is only restored in the presence of zinc
L130-133: Again, the original reports on this are not included (reference 74?). Reference 98 should be quoted here.
L134-136: there is disagreement concerning this statement; there is clear evidence that formation of a PGRMC1 and mPRα complex is needed for PRG binding and PRG functions. Aizen et al. 2018. Gen .Comp. Endocrinol. 263, 51-61. Reference 99. This statement also conflicts with a later section of this review “mPRs and PGRMC1 can form complexes”
Section on differential dynamics… : The discussion on MIF with regard to mPRs is irrelevant , MIF does not active mPRs. This entire section discusses mainly nPRs and does not add any useful information on mPRs and the entire paragraph should be deleted.
L152-159: The text here is misleading “PRG and MIF can act as negative regulators of proliferation via mPRα” MIF does not bind or activate mPRα (reference 112). Reference 88 shows MIF does not act through mPRα.Reference 89 mainly refers to nPRs. Refer 90 is a general review and refence 91 refers to PGRMC1. Reference 92 is interesting with results in nPR-ve cell lines, but does not indicate it acts through mPRα. PGRMC1 and the GR are also potential targets. This needs to be clarified.
L157-160: there is no direct evidence to support the synergistic MIF-PRG action is “solely mPR-dependent”
L161-172: This section is incomplete. Reference 73 on granulosa cells supports this association. Also, another protein. Another mPR, beta has also been shown to associate with APPL and VDL ,Nader et la., Plos Biol 2021,19,e30001117.
L184: the results in reference 108 require clarification to confirm the statement that GABA A receptors are not present in the brain ECs.
L187-198: This section suffers from not referencing the primary literature that established early on that mPRs are not GPCRS and are PAQRs. Tang et al. 2005. J.Mol.Evol 61,372-380 showed 2 years after the discovery of mPRs that they are PAQRS and reference 112 showed 2 years later that GPCRs and mPRs have a different phylogenetic origin. The authors should replace recent references 958,109) with these and revise the text. There are no reports in the primary literature that mPRs are GPCRs. EST is not defined.
Crosstalk and reciprocal regulation ……
This section reports a novel mechanism which requires substantial elaboration on the role of CCM signaling complexes in mPR functions in order to convince the reader of their importance.
L203-204: add reference 118 (118-125
L204-205: the is no evidence of coupled mPR-PR in references 118, 128 (existence of nPR in T lymphocytes disputed)
L211: The statement “nPRs, mPRs, and their shared ligands (PRG, MIF)” is incorrect. There are no data showing that MF is a ligand for mPRs. It is frequently used as a negative control, to exclude actions through the nPR. This is a major problem that requires extensive explanation of the evidence supporting this claim.
L220: reference 135 refers to sperm, not the myometrium
mPR subcellular compartmentation
This section has numerous misleading statements and requires substantial modification. The discussion about earlier controversies is not relevant and dated. There is clear evidence that mPRs are localized in intracellular regions in addition to on the cell membrane and the authors describe the evidence in lines 256-267 supporting these other localizations which is not surprising, all membrane receptors also are present intracellularly.
L236-240. The beginning of this section is redundant since it well known that all membrane receptors are also present intracellularly where they a synthesized and internalized after activation. mPRα is rapidly internalized after ligand binding : Foster et al. 2010 Mol. Med. Rep. 3,27-35. Reference 46 is a very brief review and reference 136 is a commentary over a decade ago which is no longer relevant, especially the statement L239-240 that ”debates on their subcellular compartmentation continue”. This opinion was stated 13 years ago!.
L243: the statement : “recent evidence of cytoplasmic localization ….of mPRs…have emerged” is inaccurate because reference 110 refers to a study published in 2006, and reference 140 to a review published in 2008; not recent at all.
L250: Similarly, references 141 and 142 are not “recently contradictory evidence “
L255-256 the statement “or misclassification of mPRs as membrane proteins {136,142] are still under debate” is incorrect. These papers/ commentaries were published in 2010 and 2008 and have not been seriously considered recently because of the abundant evidence the mPRs are on the cell surface and are activated by PRG -BSA ligands that cannot enter into the cell.
L270- sentence incomplete “cytoplasmic-“
272-279 there is no reference to the study discussed here.
L281-286: PGRMC1 is not a close sibling of mPRs-completely different. Why is this evidence convincing that mPRs ac the same way? This text needs to be revised.
L286-290: this section is highly speculative and not supported by any data it is suggested that it be revised to reflect this
Figure 1:
1. What is GPCRs? mPRs are not GPCRs
2. mPR are definitely on the cell surface as well as intracellularly. mPR should be shown on the cell surface.
3. Here and elsewhere in the review MIF is shown to act on mPRs. If the authors suggest MIF does not have an inhibitory effect on mPRs like it does on nPRs then the results should be discussed as such and not that MIF and PRG act together on mPRs.
This manuscript is a short review on membrane progesterone receptors (mPRs) which have been recently shown to exert numerous new functions and therefore a topic worthy of a new review. The comparisons between nPRs and mPRs in the review are interesting. The discussion of the recent studies by the authors on CCM signaling complexes coupling to both nPR and mPRs signaling cascades is particularly interesting and the section on this topic should be expanded to discuss their findings in more detail. However, overall, the scholarship is inadequate in other sections of the review. The authors have an incomplete understanding of mPRs and the text has numerous errors and misconceptions which need to be corrected. In particular, there is an over reliance on citing broad reviews on the topic written by persons not directly involved with research in mPRs. Researching the primary literature on mPRs should address many of the misconceptions and mistakes in this manuscript. The lack of web addresses (DOI) for the cited papers greatly complicates the refereeing (157 references) so detailed comments on the appropriateness of the citations are limited to those in the first 100 lines of this review. The authors should check all the citations in the rest of the review. Throughout the review authors do not always distinguish between the mPR subtypes and describe which one is being referred to. This is very important for interpreting the findings. Also, the author’s results reviewed in this manuscript indicating progesterone actions mediated solely through mPRs did not describe the use of a specific mPR agonist (Org OD 02-0) or a nPR specific one such as R5020 which would have supported their conclusions. Throughout the manuscript the authors suggest MIF and PRG act together on mPRs, although no evidence is provided to support this assertion. See comments for figure 1 for a possible alternative interpretation.. Finally, the authors make bold claims in this review of the functions on mPRs intracellularly in the absence of definitive data supporting their claims. It is strongly suggested that the authors tone down these claims and instead suggest them as hypotheses that require further examination and indicate what experiments could be conducted to test them
Review of first 100 lines
Title: The title is confusing. in some publications PGRMCs have been called mPRs e.g. Lin et al., 2015 Toxicology and Pharmacology 288 (3) 359-373. To avoid confusion, it is suggested that “(mPRs)” be replaced with “(mPRs, PAQRs)” Similarly, see the confusion in lines 68-73.
Abstract: the abstract is not very informative. The first part is repetitive and the abstract only briefly mentions the topic in the title “ are going beyond its initial definitions” It is recommended that more details on mPRs activating nPRs.
L49: The statement is incorrect and needs to be revised. It has been known since the late 1970s-1980s that progesterone acts at the cell membrane to mediate non-classical or non-genomic actions. e.g. Masui and Markert, 1971 J. Exp. Zool. 177) 129–145; Sadler and Maller 1982 J. Biol. Chem. 257 (1982) 355–361.
L52-53: This statement is unclear, what is being referred to with much earlier discovery? At this point the identities of membrane receptors have not been described, reference 29 appears to be inappropriate here, does not directly refer to this statement.
L56: here with reference 30 and elsewhere in this review the authors rely on reviews rather than the primary literature. This reference is especially inappropriate since it refers mainly to PGRMC1. The combined responses should be briefly listed here. For example, a recent example of a combined response through nPRs and mPRs : Pang and Thomas, 2022, J. Mol. Endocrinol. 70:1 e220073.
L 65: Title is incorrect see comment about line 49 above . It was known from early times that mPRs act independently to nPRs.
L68-73 and L81-82.:The term mPRs is confusing as discussed earlier for the title. Change mPRs to membrane progesterone receptors, the term mPRs should be reserved for PAQRs in this review.
L80: it should be clarified that PGRMC1 only has moderate binding affinity for PRG.
L84 Reference 44 there are better references than this such as 125,126
45. should include original reference for activating G proteins (reference 112)
L87-88: the statement is incorrect because the reference referred to here is a short meeting abstract with preliminary results which were inconclusive. A more complete study by Wu, Liu, Chen, Hong and Zhu 2020 Mol Cell Endocrinol. 511: 110856 shows that there are differential effects of knockout of the different mPRs on oocyte maturation and ovulation. The next needs to be revised accordingly.
L94: reference 34 is not appropriate here. These authors primarily discuss extranuclear signaling through the nPR. The paper by Ashley et al.,2006 Endocrinology 147 (9) 4145-4159 describes a sheep mPR and its localization in reproductive tissues including ovary.
L98: references 56,57 do not refer to mPRs in the CNS. More appropriate references are Meffre et al. 2013 Neuroscience 231: 111-124, Zuloaga et al 2012 Endocrinology 153:4432-443.
Abbreviated Review of rest of manuscript.
L104-109: The statement is unclear and needs to be revised. It is not clear that arginine is not in the binding pocket of wild type mPRα, only when its substituted at a key residue; this destroys binding which is only restored in the presence of zinc
L130-133: Again, the original reports on this are not included (reference 74?). Reference 98 should be quoted here.
L134-136: there is disagreement concerning this statement; there is clear evidence that formation of a PGRMC1 and mPRα complex is needed for PRG binding and PRG functions. Aizen et al. 2018. Gen .Comp. Endocrinol. 263, 51-61. Reference 99. This statement also conflicts with a later section of this review “mPRs and PGRMC1 can form complexes”
Section on differential dynamics… : The discussion on MIF with regard to mPRs is irrelevant , MIF does not active mPRs. This entire section discusses mainly nPRs and does not add any useful information on mPRs and the entire paragraph should be deleted.
L152-159: The text here is misleading “PRG and MIF can act as negative regulators of proliferation via mPRα” MIF does not bind or activate mPRα (reference 112). Reference 88 shows MIF does not act through mPRα.Reference 89 mainly refers to nPRs. Refer 90 is a general review and refence 91 refers to PGRMC1. Reference 92 is interesting with results in nPR-ve cell lines, but does not indicate it acts through mPRα. PGRMC1 and the GR are also potential targets. This needs to be clarified.
L157-160: there is no direct evidence to support the synergistic MIF-PRG action is “solely mPR-dependent”
L161-172: This section is incomplete. Reference 73 on granulosa cells supports this association. Also, another protein. Another mPR, beta has also been shown to associate with APPL and VDL ,Nader et la., Plos Biol 2021,19,e30001117.
L184: the results in reference 108 require clarification to confirm the statement that GABA A receptors are not present in the brain ECs.
L187-198: This section suffers from not referencing the primary literature that established early on that mPRs are not GPCRS and are PAQRs. Tang et al. 2005. J.Mol.Evol 61,372-380 showed 2 years after the discovery of mPRs that they are PAQRS and reference 112 showed 2 years later that GPCRs and mPRs have a different phylogenetic origin. The authors should replace recent references 958,109) with these and revise the text. There are no reports in the primary literature that mPRs are GPCRs. EST is not defined.
Crosstalk and reciprocal regulation ……
This section reports a novel mechanism which requires substantial elaboration on the role of CCM signaling complexes in mPR functions in order to convince the reader of their importance.
L203-204: add reference 118 (118-125
L204-205: the is no evidence of coupled mPR-PR in references 118, 128 (existence of nPR in T lymphocytes disputed)
L211: The statement “nPRs, mPRs, and their shared ligands (PRG, MIF)” is incorrect. There are no data showing that MF is a ligand for mPRs. It is frequently used as a negative control, to exclude actions through the nPR. This is a major problem that requires extensive explanation of the evidence supporting this claim.
L220: reference 135 refers to sperm, not the myometrium
mPR subcellular compartmentation
This section has numerous misleading statements and requires substantial modification. The discussion about earlier controversies is not relevant and dated. There is clear evidence that mPRs are localized in intracellular regions in addition to on the cell membrane and the authors describe the evidence in lines 256-267 supporting these other localizations which is not surprising, all membrane receptors also are present intracellularly.
L236-240. The beginning of this section is redundant since it well known that all membrane receptors are also present intracellularly where they a synthesized and internalized after activation. mPRα is rapidly internalized after ligand binding : Foster et al. 2010 Mol. Med. Rep. 3,27-35. Reference 46 is a very brief review and reference 136 is a commentary over a decade ago which is no longer relevant, especially the statement L239-240 that ”debates on their subcellular compartmentation continue”. This opinion was stated 13 years ago!.
L243: the statement : “recent evidence of cytoplasmic localization ….of mPRs…have emerged” is inaccurate because reference 110 refers to a study published in 2006, and reference 140 to a review published in 2008; not recent at all.
L250: Similarly, references 141 and 142 are not “recently contradictory evidence “
L255-256 the statement “or misclassification of mPRs as membrane proteins {136,142] are still under debate” is incorrect. These papers/ commentaries were published in 2010 and 2008 and have not been seriously considered recently because of the abundant evidence the mPRs are on the cell surface and are activated by PRG -BSA ligands that cannot enter into the cell.
L270- sentence incomplete “cytoplasmic-“
272-279 there is no reference to the study discussed here.
L281-286: PGRMC1 is not a close sibling of mPRs-completely different. Why is this evidence convincing that mPRs ac the same way? This text needs to be revised.
L286-290: this section is highly speculative and not supported by any data it is suggested that it be revised to reflect this
Figure 1:
1. What is GPCRs? mPRs are not GPCRs
2. mPR are definitely on the cell surface as well as intracellularly. mPR should be shown on the cell surface.
3. Here and elsewhere in the review MIF is shown to act on mPRs. If the authors suggest MIF does not have an inhibitory effect on mPRs like it does on nPRs then the results should be discussed as such and not that MIF and PRG act together on mPRs.
This manuscript is a short review on membrane progesterone receptors (mPRs) which have been recently shown to exert numerous new functions and therefore a topic worthy of a new review. The comparisons between nPRs and mPRs in the review are interesting. The discussion of the recent studies by the authors on CCM signaling complexes coupling to both nPR and mPRs signaling cascades is particularly interesting and the section on this topic should be expanded to discuss their findings in more detail. However, overall, the scholarship is inadequate in other sections of the review. The authors have an incomplete understanding of mPRs and the text has numerous errors and misconceptions which need to be corrected. In particular, there is an over reliance on citing broad reviews on the topic written by persons not directly involved with research in mPRs. Researching the primary literature on mPRs should address many of the misconceptions and mistakes in this manuscript. The lack of web addresses (DOI) for the cited papers greatly complicates the refereeing (157 references) so detailed comments on the appropriateness of the citations are limited to those in the first 100 lines of this review. The authors should check all the citations in the rest of the review. Throughout the review authors do not always distinguish between the mPR subtypes and describe which one is being referred to. This is very important for interpreting the findings. Also, the author’s results reviewed in this manuscript indicating progesterone actions mediated solely through mPRs did not describe the use of a specific mPR agonist (Org OD 02-0) or a nPR specific one such as R5020 which would have supported their conclusions. Throughout the manuscript the authors suggest MIF and PRG act together on mPRs, although no evidence is provided to support this assertion. See comments for figure 1 for a possible alternative interpretation.. Finally, the authors make bold claims in this review of the functions on mPRs intracellularly in the absence of definitive data supporting their claims. It is strongly suggested that the authors tone down these claims and instead suggest them as hypotheses that require further examination and indicate what experiments could be conducted to test them
Review of first 100 lines
Title: The title is confusing. in some publications PGRMCs have been called mPRs e.g. Lin et al., 2015 Toxicology and Pharmacology 288 (3) 359-373. To avoid confusion, it is suggested that “(mPRs)” be replaced with “(mPRs, PAQRs)” Similarly, see the confusion in lines 68-73.
Abstract: the abstract is not very informative. The first part is repetitive and the abstract only briefly mentions the topic in the title “ are going beyond its initial definitions” It is recommended that more details on mPRs activating nPRs.
L49: The statement is incorrect and needs to be revised. It has been known since the late 1970s-1980s that progesterone acts at the cell membrane to mediate non-classical or non-genomic actions. e.g. Masui and Markert, 1971 J. Exp. Zool. 177) 129–145; Sadler and Maller 1982 J. Biol. Chem. 257 (1982) 355–361.
L52-53: This statement is unclear, what is being referred to with much earlier discovery? At this point the identities of membrane receptors have not been described, reference 29 appears to be inappropriate here, does not directly refer to this statement.
L56: here with reference 30 and elsewhere in this review the authors rely on reviews rather than the primary literature. This reference is especially inappropriate since it refers mainly to PGRMC1. The combined responses should be briefly listed here. For example, a recent example of a combined response through nPRs and mPRs : Pang and Thomas, 2022, J. Mol. Endocrinol. 70:1 e220073.
L 65: Title is incorrect see comment about line 49 above . It was known from early times that mPRs act independently to nPRs.
L68-73 and L81-82.:The term mPRs is confusing as discussed earlier for the title. Change mPRs to membrane progesterone receptors, the term mPRs should be reserved for PAQRs in this review.
L80: it should be clarified that PGRMC1 only has moderate binding affinity for PRG.
L84 Reference 44 there are better references than this such as 125,126
45. should include original reference for activating G proteins (reference 112)
L87-88: the statement is incorrect because the reference referred to here is a short meeting abstract with preliminary results which were inconclusive. A more complete study by Wu, Liu, Chen, Hong and Zhu 2020 Mol Cell Endocrinol. 511: 110856 shows that there are differential effects of knockout of the different mPRs on oocyte maturation and ovulation. The next needs to be revised accordingly.
L94: reference 34 is not appropriate here. These authors primarily discuss extranuclear signaling through the nPR. The paper by Ashley et al.,2006 Endocrinology 147 (9) 4145-4159 describes a sheep mPR and its localization in reproductive tissues including ovary.
L98: references 56,57 do not refer to mPRs in the CNS. More appropriate references are Meffre et al. 2013 Neuroscience 231: 111-124, Zuloaga et al 2012 Endocrinology 153:4432-443.
Abbreviated Review of rest of manuscript.
L104-109: The statement is unclear and needs to be revised. It is not clear that arginine is not in the binding pocket of wild type mPRα, only when its substituted at a key residue; this destroys binding which is only restored in the presence of zinc
L130-133: Again, the original reports on this are not included (reference 74?). Reference 98 should be quoted here.
L134-136: there is disagreement concerning this statement; there is clear evidence that formation of a PGRMC1 and mPRα complex is needed for PRG binding and PRG functions. Aizen et al. 2018. Gen .Comp. Endocrinol. 263, 51-61. Reference 99. This statement also conflicts with a later section of this review “mPRs and PGRMC1 can form complexes”
Section on differential dynamics… : The discussion on MIF with regard to mPRs is irrelevant , MIF does not active mPRs. This entire section discusses mainly nPRs and does not add any useful information on mPRs and the entire paragraph should be deleted.
L152-159: The text here is misleading “PRG and MIF can act as negative regulators of proliferation via mPRα” MIF does not bind or activate mPRα (reference 112). Reference 88 shows MIF does not act through mPRα.Reference 89 mainly refers to nPRs. Refer 90 is a general review and refence 91 refers to PGRMC1. Reference 92 is interesting with results in nPR-ve cell lines, but does not indicate it acts through mPRα. PGRMC1 and the GR are also potential targets. This needs to be clarified.
L157-160: there is no direct evidence to support the synergistic MIF-PRG action is “solely mPR-dependent”
L161-172: This section is incomplete. Reference 73 on granulosa cells supports this association. Also, another protein. Another mPR, beta has also been shown to associate with APPL and VDL ,Nader et la., Plos Biol 2021,19,e30001117.
L184: the results in reference 108 require clarification to confirm the statement that GABA A receptors are not present in the brain ECs.
L187-198: This section suffers from not referencing the primary literature that established early on that mPRs are not GPCRS and are PAQRs. Tang et al. 2005. J.Mol.Evol 61,372-380 showed 2 years after the discovery of mPRs that they are PAQRS and reference 112 showed 2 years later that GPCRs and mPRs have a different phylogenetic origin. The authors should replace recent references 958,109) with these and revise the text. There are no reports in the primary literature that mPRs are GPCRs. EST is not defined.
Crosstalk and reciprocal regulation ……
This section reports a novel mechanism which requires substantial elaboration on the role of CCM signaling complexes in mPR functions in order to convince the reader of their importance.
L203-204: add reference 118 (118-125
L204-205: the is no evidence of coupled mPR-PR in references 118, 128 (existence of nPR in T lymphocytes disputed)
L211: The statement “nPRs, mPRs, and their shared ligands (PRG, MIF)” is incorrect. There are no data showing that MF is a ligand for mPRs. It is frequently used as a negative control, to exclude actions through the nPR. This is a major problem that requires extensive explanation of the evidence supporting this claim.
L220: reference 135 refers to sperm, not the myometrium
mPR subcellular compartmentation
This section has numerous misleading statements and requires substantial modification. The discussion about earlier controversies is not relevant and dated. There is clear evidence that mPRs are localized in intracellular regions in addition to on the cell membrane and the authors describe the evidence in lines 256-267 supporting these other localizations which is not surprising, all membrane receptors also are present intracellularly.
L236-240. The beginning of this section is redundant since it well known that all membrane receptors are also present intracellularly where they a synthesized and internalized after activation. mPRα is rapidly internalized after ligand binding : Foster et al. 2010 Mol. Med. Rep. 3,27-35. Reference 46 is a very brief review and reference 136 is a commentary over a decade ago which is no longer relevant, especially the statement L239-240 that ”debates on their subcellular compartmentation continue”. This opinion was stated 13 years ago!.
L243: the statement : “recent evidence of cytoplasmic localization ….of mPRs…have emerged” is inaccurate because reference 110 refers to a study published in 2006, and reference 140 to a review published in 2008; not recent at all.
L250: Similarly, references 141 and 142 are not “recently contradictory evidence “
L255-256 the statement “or misclassification of mPRs as membrane proteins {136,142] are still under debate” is incorrect. These papers/ commentaries were published in 2010 and 2008 and have not been seriously considered recently because of the abundant evidence the mPRs are on the cell surface and are activated by PRG -BSA ligands that cannot enter into the cell.
L270- sentence incomplete “cytoplasmic-“
272-279 there is no reference to the study discussed here.
L281-286: PGRMC1 is not a close sibling of mPRs-completely different. Why is this evidence convincing that mPRs ac the same way? This text needs to be revised.
L286-290: this section is highly speculative and not supported by any data it is suggested that it be revised to reflect this
Figure 1:
1. What is GPCRs? mPRs are not GPCRs
2. mPR are definitely on the cell surface as well as intracellularly. mPR should be shown on the cell surface.
3. Here and elsewhere in the review MIF is shown to act on mPRs. If the authors suggest MIF does not have an inhibitory effect on mPRs like it does on nPRs then the results should be discussed as such and not that MIF and PRG act together on mPRs.
This manuscript is a short review on membrane progesterone receptors (mPRs) which have been recently shown to exert numerous new functions and therefore a topic worthy of a new review. The comparisons between nPRs and mPRs in the review are interesting. The discussion of the recent studies by the authors on CCM signaling complexes coupling to both nPR and mPRs signaling cascades is particularly interesting and the section on this topic should be expanded to discuss their findings in more detail. However, overall, the scholarship is inadequate in other sections of the review. The authors have an incomplete understanding of mPRs and the text has numerous errors and misconceptions which need to be corrected. In particular, there is an over reliance on citing broad reviews on the topic written by persons not directly involved with research in mPRs. Researching the primary literature on mPRs should address many of the misconceptions and mistakes in this manuscript. The lack of web addresses (DOI) for the cited papers greatly complicates the refereeing (157 references) so detailed comments on the appropriateness of the citations are limited to those in the first 100 lines of this review. The authors should check all the citations in the rest of the review. Throughout the review authors do not always distinguish between the mPR subtypes and describe which one is being referred to. This is very important for interpreting the findings. Also, the author’s results reviewed in this manuscript indicating progesterone actions mediated solely through mPRs did not describe the use of a specific mPR agonist (Org OD 02-0) or a nPR specific one such as R5020 which would have supported their conclusions. Throughout the manuscript the authors suggest MIF and PRG act together on mPRs, although no evidence is provided to support this assertion. See comments for figure 1 for a possible alternative interpretation.. Finally, the authors make bold claims in this review of the functions on mPRs intracellularly in the absence of definitive data supporting their claims. It is strongly suggested that the authors tone down these claims and instead suggest them as hypotheses that require further examination and indicate what experiments could be conducted to test them
Review of first 100 lines
Title: The title is confusing. in some publications PGRMCs have been called mPRs e.g. Lin et al., 2015 Toxicology and Pharmacology 288 (3) 359-373. To avoid confusion, it is suggested that “(mPRs)” be replaced with “(mPRs, PAQRs)” Similarly, see the confusion in lines 68-73.
Abstract: the abstract is not very informative. The first part is repetitive and the abstract only briefly mentions the topic in the title “ are going beyond its initial definitions” It is recommended that more details on mPRs activating nPRs.
L49: The statement is incorrect and needs to be revised. It has been known since the late 1970s-1980s that progesterone acts at the cell membrane to mediate non-classical or non-genomic actions. e.g. Masui and Markert, 1971 J. Exp. Zool. 177) 129–145; Sadler and Maller 1982 J. Biol. Chem. 257 (1982) 355–361.
L52-53: This statement is unclear, what is being referred to with much earlier discovery? At this point the identities of membrane receptors have not been described, reference 29 appears to be inappropriate here, does not directly refer to this statement.
L56: here with reference 30 and elsewhere in this review the authors rely on reviews rather than the primary literature. This reference is especially inappropriate since it refers mainly to PGRMC1. The combined responses should be briefly listed here. For example, a recent example of a combined response through nPRs and mPRs : Pang and Thomas, 2022, J. Mol. Endocrinol. 70:1 e220073.
L 65: Title is incorrect see comment about line 49 above . It was known from early times that mPRs act independently to nPRs.
L68-73 and L81-82.:The term mPRs is confusing as discussed earlier for the title. Change mPRs to membrane progesterone receptors, the term mPRs should be reserved for PAQRs in this review.
L80: it should be clarified that PGRMC1 only has moderate binding affinity for PRG.
L84 Reference 44 there are better references than this such as 125,126
45. should include original reference for activating G proteins (reference 112)
L87-88: the statement is incorrect because the reference referred to here is a short meeting abstract with preliminary results which were inconclusive. A more complete study by Wu, Liu, Chen, Hong and Zhu 2020 Mol Cell Endocrinol. 511: 110856 shows that there are differential effects of knockout of the different mPRs on oocyte maturation and ovulation. The next needs to be revised accordingly.
L94: reference 34 is not appropriate here. These authors primarily discuss extranuclear signaling through the nPR. The paper by Ashley et al.,2006 Endocrinology 147 (9) 4145-4159 describes a sheep mPR and its localization in reproductive tissues including ovary.
L98: references 56,57 do not refer to mPRs in the CNS. More appropriate references are Meffre et al. 2013 Neuroscience 231: 111-124, Zuloaga et al 2012 Endocrinology 153:4432-443.
Abbreviated Review of rest of manuscript.
L104-109: The statement is unclear and needs to be revised. It is not clear that arginine is not in the binding pocket of wild type mPRα, only when its substituted at a key residue; this destroys binding which is only restored in the presence of zinc
L130-133: Again, the original reports on this are not included (reference 74?). Reference 98 should be quoted here.
L134-136: there is disagreement concerning this statement; there is clear evidence that formation of a PGRMC1 and mPRα complex is needed for PRG binding and PRG functions. Aizen et al. 2018. Gen .Comp. Endocrinol. 263, 51-61. Reference 99. This statement also conflicts with a later section of this review “mPRs and PGRMC1 can form complexes”
Section on differential dynamics… : The discussion on MIF with regard to mPRs is irrelevant , MIF does not active mPRs. This entire section discusses mainly nPRs and does not add any useful information on mPRs and the entire paragraph should be deleted.
L152-159: The text here is misleading “PRG and MIF can act as negative regulators of proliferation via mPRα” MIF does not bind or activate mPRα (reference 112). Reference 88 shows MIF does not act through mPRα.Reference 89 mainly refers to nPRs. Refer 90 is a general review and refence 91 refers to PGRMC1. Reference 92 is interesting with results in nPR-ve cell lines, but does not indicate it acts through mPRα. PGRMC1 and the GR are also potential targets. This needs to be clarified.
L157-160: there is no direct evidence to support the synergistic MIF-PRG action is “solely mPR-dependent”
L161-172: This section is incomplete. Reference 73 on granulosa cells supports this association. Also, another protein. Another mPR, beta has also been shown to associate with APPL and VDL ,Nader et la., Plos Biol 2021,19,e30001117.
L184: the results in reference 108 require clarification to confirm the statement that GABA A receptors are not present in the brain ECs.
L187-198: This section suffers from not referencing the primary literature that established early on that mPRs are not GPCRS and are PAQRs. Tang et al. 2005. J.Mol.Evol 61,372-380 showed 2 years after the discovery of mPRs that they are PAQRS and reference 112 showed 2 years later that GPCRs and mPRs have a different phylogenetic origin. The authors should replace recent references 958,109) with these and revise the text. There are no reports in the primary literature that mPRs are GPCRs. EST is not defined.
Crosstalk and reciprocal regulation ……
This section reports a novel mechanism which requires substantial elaboration on the role of CCM signaling complexes in mPR functions in order to convince the reader of their importance.
L203-204: add reference 118 (118-125
L204-205: the is no evidence of coupled mPR-PR in references 118, 128 (existence of nPR in T lymphocytes disputed)
L211: The statement “nPRs, mPRs, and their shared ligands (PRG, MIF)” is incorrect. There are no data showing that MF is a ligand for mPRs. It is frequently used as a negative control, to exclude actions through the nPR. This is a major problem that requires extensive explanation of the evidence supporting this claim.
L220: reference 135 refers to sperm, not the myometrium
mPR subcellular compartmentation
This section has numerous misleading statements and requires substantial modification. The discussion about earlier controversies is not relevant and dated. There is clear evidence that mPRs are localized in intracellular regions in addition to on the cell membrane and the authors describe the evidence in lines 256-267 supporting these other localizations which is not surprising, all membrane receptors also are present intracellularly.
L236-240. The beginning of this section is redundant since it well known that all membrane receptors are also present intracellularly where they a synthesized and internalized after activation. mPRα is rapidly internalized after ligand binding : Foster et al. 2010 Mol. Med. Rep. 3,27-35. Reference 46 is a very brief review and reference 136 is a commentary over a decade ago which is no longer relevant, especially the statement L239-240 that ”debates on their subcellular compartmentation continue”. This opinion was stated 13 years ago!.
L243: the statement : “recent evidence of cytoplasmic localization ….of mPRs…have emerged” is inaccurate because reference 110 refers to a study published in 2006, and reference 140 to a review published in 2008; not recent at all.
L250: Similarly, references 141 and 142 are not “recently contradictory evidence “
L255-256 the statement “or misclassification of mPRs as membrane proteins {136,142] are still under debate” is incorrect. These papers/ commentaries were published in 2010 and 2008 and have not been seriously considered recently because of the abundant evidence the mPRs are on the cell surface and are activated by PRG -BSA ligands that cannot enter into the cell.
L270- sentence incomplete “cytoplasmic-“
272-279 there is no reference to the study discussed here.
L281-286: PGRMC1 is not a close sibling of mPRs-completely different. Why is this evidence convincing that mPRs ac the same way? This text needs to be revised.
L286-290: this section is highly speculative and not supported by any data it is suggested that it be revised to reflect this
Figure 1:
1. What is GPCRs? mPRs are not GPCRs
2. mPR are definitely on the cell surface as well as intracellularly. mPR should be shown on the cell surface.
3. Here and elsewhere in the review MIF is shown to act on mPRs. If the authors suggest MIF does not have an inhibitory effect on mPRs like it does on nPRs then the results should be discussed as such and not that MIF and PRG act together on mPRs.
This manuscript is a short review on membrane progesterone receptors (mPRs) which have been recently shown to exert numerous new functions and therefore a topic worthy of a new review. The comparisons between nPRs and mPRs in the review are interesting. The discussion of the recent studies by the authors on CCM signaling complexes coupling to both nPR and mPRs signaling cascades is particularly interesting and the section on this topic should be expanded to discuss their findings in more detail. However, overall, the scholarship is inadequate in other sections of the review. The authors have an incomplete understanding of mPRs and the text has numerous errors and misconceptions which need to be corrected. In particular, there is an over reliance on citing broad reviews on the topic written by persons not directly involved with research in mPRs. Researching the primary literature on mPRs should address many of the misconceptions and mistakes in this manuscript. The lack of web addresses (DOI) for the cited papers greatly complicates the refereeing (157 references) so detailed comments on the appropriateness of the citations are limited to those in the first 100 lines of this review. The authors should check all the citations in the rest of the review. Throughout the review authors do not always distinguish between the mPR subtypes and describe which one is being referred to. This is very important for interpreting the findings. Also, the author’s results reviewed in this manuscript indicating progesterone actions mediated solely through mPRs did not describe the use of a specific mPR agonist (Org OD 02-0) or a nPR specific one such as R5020 which would have supported their conclusions. Throughout the manuscript the authors suggest MIF and PRG act together on mPRs, although no evidence is provided to support this assertion. See comments for figure 1 for a possible alternative interpretation.. Finally, the authors make bold claims in this review of the functions on mPRs intracellularly in the absence of definitive data supporting their claims. It is strongly suggested that the authors tone down these claims and instead suggest them as hypotheses that require further examination and indicate what experiments could be conducted to test them
Review of first 100 lines
Title: The title is confusing. in some publications PGRMCs have been called mPRs e.g. Lin et al., 2015 Toxicology and Pharmacology 288 (3) 359-373. To avoid confusion, it is suggested that “(mPRs)” be replaced with “(mPRs, PAQRs)” Similarly, see the confusion in lines 68-73.
Abstract: the abstract is not very informative. The first part is repetitive and the abstract only briefly mentions the topic in the title “ are going beyond its initial definitions” It is recommended that more details on mPRs activating nPRs.
L49: The statement is incorrect and needs to be revised. It has been known since the late 1970s-1980s that progesterone acts at the cell membrane to mediate non-classical or non-genomic actions. e.g. Masui and Markert, 1971 J. Exp. Zool. 177) 129–145; Sadler and Maller 1982 J. Biol. Chem. 257 (1982) 355–361.
L52-53: This statement is unclear, what is being referred to with much earlier discovery? At this point the identities of membrane receptors have not been described, reference 29 appears to be inappropriate here, does not directly refer to this statement.
L56: here with reference 30 and elsewhere in this review the authors rely on reviews rather than the primary literature. This reference is especially inappropriate since it refers mainly to PGRMC1. The combined responses should be briefly listed here. For example, a recent example of a combined response through nPRs and mPRs : Pang and Thomas, 2022, J. Mol. Endocrinol. 70:1 e220073.
L 65: Title is incorrect see comment about line 49 above . It was known from early times that mPRs act independently to nPRs.
L68-73 and L81-82.:The term mPRs is confusing as discussed earlier for the title. Change mPRs to membrane progesterone receptors, the term mPRs should be reserved for PAQRs in this review.
L80: it should be clarified that PGRMC1 only has moderate binding affinity for PRG.
L84 Reference 44 there are better references than this such as 125,126
45. should include original reference for activating G proteins (reference 112)
L87-88: the statement is incorrect because the reference referred to here is a short meeting abstract with preliminary results which were inconclusive. A more complete study by Wu, Liu, Chen, Hong and Zhu 2020 Mol Cell Endocrinol. 511: 110856 shows that there are differential effects of knockout of the different mPRs on oocyte maturation and ovulation. The next needs to be revised accordingly.
L94: reference 34 is not appropriate here. These authors primarily discuss extranuclear signaling through the nPR. The paper by Ashley et al.,2006 Endocrinology 147 (9) 4145-4159 describes a sheep mPR and its localization in reproductive tissues including ovary.
L98: references 56,57 do not refer to mPRs in the CNS. More appropriate references are Meffre et al. 2013 Neuroscience 231: 111-124, Zuloaga et al 2012 Endocrinology 153:4432-443.
Abbreviated Review of rest of manuscript.
L104-109: The statement is unclear and needs to be revised. It is not clear that arginine is not in the binding pocket of wild type mPRα, only when its substituted at a key residue; this destroys binding which is only restored in the presence of zinc
L130-133: Again, the original reports on this are not included (reference 74?). Reference 98 should be quoted here.
L134-136: there is disagreement concerning this statement; there is clear evidence that formation of a PGRMC1 and mPRα complex is needed for PRG binding and PRG functions. Aizen et al. 2018. Gen .Comp. Endocrinol. 263, 51-61. Reference 99. This statement also conflicts with a later section of this review “mPRs and PGRMC1 can form complexes”
Section on differential dynamics… : The discussion on MIF with regard to mPRs is irrelevant , MIF does not active mPRs. This entire section discusses mainly nPRs and does not add any useful information on mPRs and the entire paragraph should be deleted.
L152-159: The text here is misleading “PRG and MIF can act as negative regulators of proliferation via mPRα” MIF does not bind or activate mPRα (reference 112). Reference 88 shows MIF does not act through mPRα.Reference 89 mainly refers to nPRs. Refer 90 is a general review and refence 91 refers to PGRMC1. Reference 92 is interesting with results in nPR-ve cell lines, but does not indicate it acts through mPRα. PGRMC1 and the GR are also potential targets. This needs to be clarified.
L157-160: there is no direct evidence to support the synergistic MIF-PRG action is “solely mPR-dependent”
L161-172: This section is incomplete. Reference 73 on granulosa cells supports this association. Also, another protein. Another mPR, beta has also been shown to associate with APPL and VDL ,Nader et la., Plos Biol 2021,19,e30001117.
L184: the results in reference 108 require clarification to confirm the statement that GABA A receptors are not present in the brain ECs.
L187-198: This section suffers from not referencing the primary literature that established early on that mPRs are not GPCRS and are PAQRs. Tang et al. 2005. J.Mol.Evol 61,372-380 showed 2 years after the discovery of mPRs that they are PAQRS and reference 112 showed 2 years later that GPCRs and mPRs have a different phylogenetic origin. The authors should replace recent references 958,109) with these and revise the text. There are no reports in the primary literature that mPRs are GPCRs. EST is not defined.
Crosstalk and reciprocal regulation ……
This section reports a novel mechanism which requires substantial elaboration on the role of CCM signaling complexes in mPR functions in order to convince the reader of their importance.
L203-204: add reference 118 (118-125
L204-205: the is no evidence of coupled mPR-PR in references 118, 128 (existence of nPR in T lymphocytes disputed)
L211: The statement “nPRs, mPRs, and their shared ligands (PRG, MIF)” is incorrect. There are no data showing that MF is a ligand for mPRs. It is frequently used as a negative control, to exclude actions through the nPR. This is a major problem that requires extensive explanation of the evidence supporting this claim.
L220: reference 135 refers to sperm, not the myometrium
mPR subcellular compartmentation
This section has numerous misleading statements and requires substantial modification. The discussion about earlier controversies is not relevant and dated. There is clear evidence that mPRs are localized in intracellular regions in addition to on the cell membrane and the authors describe the evidence in lines 256-267 supporting these other localizations which is not surprising, all membrane receptors also are present intracellularly.
L236-240. The beginning of this section is redundant since it well known that all membrane receptors are also present intracellularly where they a synthesized and internalized after activation. mPRα is rapidly internalized after ligand binding : Foster et al. 2010 Mol. Med. Rep. 3,27-35. Reference 46 is a very brief review and reference 136 is a commentary over a decade ago which is no longer relevant, especially the statement L239-240 that ”debates on their subcellular compartmentation continue”. This opinion was stated 13 years ago!.
L243: the statement : “recent evidence of cytoplasmic localization ….of mPRs…have emerged” is inaccurate because reference 110 refers to a study published in 2006, and reference 140 to a review published in 2008; not recent at all.
L250: Similarly, references 141 and 142 are not “recently contradictory evidence “
L255-256 the statement “or misclassification of mPRs as membrane proteins {136,142] are still under debate” is incorrect. These papers/ commentaries were published in 2010 and 2008 and have not been seriously considered recently because of the abundant evidence the mPRs are on the cell surface and are activated by PRG -BSA ligands that cannot enter into the cell.
L270- sentence incomplete “cytoplasmic-“
272-279 there is no reference to the study discussed here.
L281-286: PGRMC1 is not a close sibling of mPRs-completely different. Why is this evidence convincing that mPRs ac the same way? This text needs to be revised.
L286-290: this section is highly speculative and not supported by any data it is suggested that it be revised to reflect this
Figure 1:
1. What is GPCRs? mPRs are not GPCRs
2. mPR are definitely on the cell surface as well as intracellularly. mPR should be shown on the cell surface.
3. Here and elsewhere in the review MIF is shown to act on mPRs. If the authors suggest MIF does not have an inhibitory effect on mPRs like it does on nPRs then the results should be discussed as such and not that MIF and PRG act together on mPRs.
Author Response
Please see attached letter.
Thank you!
Jun

Reviewer 2 Report
The submitted review article provides a comprehensive overview of progesterone and its signaling occurring via the activation of genomic and non-genomic pathways. The manuscript is well done and I only suggest few minor queries.
Correct format edits in lanes 138, 162, 174
Number paragraphs and sub-paragraphs
Format reference list to MDPI style
Author Response

(The authors gave the same response as above.)

Reviewer 3 Report
A nice review that compiles information on progesterone receptors from their early discovery/exploration until more recent times. The articles published in 2020-2023 are presented in considerable numbers - 14, 8 and 20 respectively. The manuscript is easy to read and well paragraphed. There is a small bémol in the text, for example:
- Lines 39-40, the reference is cut in two parts;
- Lines 138; 162, 174: the title written by different character styles and some words are highlighted? Why ?
- The point in chapter/paragraphe titles is nor nessassairy
- Line 27 : ‘…for cytoplasmic’ of what?
Author Response

(The authors gave the same response as above.)

Round 2
Reviewer 1 Report
The authors have conscientiously answered all the questions raised by the reviewer and addressed many of the reviewer's concerns. The manuscript is significantly improved as a result. As noted by the authors many issues concerning the exact roles and mechanisms of mPR actions remain to be resolved and there continues to be uncertainty and controversy of their characteristics. Therefore it is expected that there is some disagreement of the interpretation of the results of some of the studies.